# Dahl Friction Model for Driving Characteristics of V-Shape Linear Ultrasonic Motors

**DOI:** 10.3390/mi13091407

**Published:** 2022-08-27

**Authors:** Bo Zhang, Xianghui Yuan, Yuansong Zeng, Lihui Lang, Hailong Liang, Yanhu Zhang

**Affiliations:** 1School of Mechanical Engineering and Automation, Beihang University, Beijing 100191, China; 2AVIC Manufacturing Technology Institute, Beijing 100024, China; 3Institute of Advanced Manufacturing and Modern Equipment Technology, Jiangsu University, 301 Xuefu Road, Zhenjiang 212013, China

**Keywords:** ultrasonic motor, standing wave, dynamic contact, friction drive, Dahl model, speed difference

## Abstract

The contact process of stator and slider described by the Coulomb friction model is basically in a pure sliding friction state, and a mechanical model based on the Dahl friction theory was proposed to describe the contact process between stator and slider of V-shape linear ultrasonic motor. With consideration for the tangential compliance of stator and slider, the dynamic contact and friction processes of stator and slider were addressed in stages. The simulation results show that the ratio of the friction positive work decreases with the increase of the preload, and the vibration amplitude of the stator increases the proportion of positive work of the friction force. Improving the contact stiffness of the stator and slider is conducive to improving the output performance of the ultrasonic motor. The asymmetry of the left and right performance of the V-shaped vibrator will cause a difference in the left and right running speeds of the ultrasonic motor. The improved Dahl friction-driving model makes up for the discontinuity of tangential contact force calculated by the Coulomb friction model. This study demonstrates that the friction-driving model based on the Dahl theory is reliable and reasonable for linear ultrasonic motors according to the experimental results.

## 1. Introduction

Linear ultrasonic motors are a type of actuation micromachine used for various precision applications [1,2], which working principle includes two classical physical processes. One is the generation of the high-frequency and micro-amplitude vibration of the elastic vibrator(s) using the inverse piezoelectric effect of piezoelectric ceramics [3,4]; the other is that the realization of linear motion of the slider or rotor is driven through the friction function of the contact couple [5,6]. Linear ultrasonic motors have attracted wide attention for their advantages [2,7,8], such as quick response, precision positioning [9], low/high speed and high thrust [4,10], free from electromagnetic interference, direct drive (without transmission parts, such as a gearbox), and power-off self-locking [11]. The linear ultrasonic motors have been adequately researched by scholars from Japan, Germany, USA, South Korea, Singapore, China, and other countries (see Figure 1 and Figure 2), covering drive modes [12], vibrator structures [13], excitation conditions (input signals [14], waveform [1,15,16], phase difference [17]), noise elimination [18], control strategy [19,20,21], performances measurement [22,23], friction materials [24,25,26] and coatings [27], damping and nonlinear material behavior [28], (non)contact behavior [29,30,31,32], driving characteristics [33,34,35], lubrication strategy [36,37,38], wear lifespan [39,40], and working environment [41]. In recent years, a variety of linear ultrasonic motors have been developed [35,42,43,44,45,46] for structure design and performance improvement [47,48]. However, research on the friction behavior and driving mechanism of linear ultrasonic motors are relatively scarce [49,50]. Adachi et al. simulated the contact behavior of the ultrasonic wave motor and measured the friction force at the contact interface between a rotational disk and an oscillatory pin induced by ultrasonic wave [51]. Kurosawa et al. investigated the operation of an ultrasonic linear motor using Rayleigh waves and pointed out that the shape of the contact surface and the slider material is critical [49]. In addition, the depression was independent to the contact pressure and independent to the projection density [52]. Nakamura and Ueha reviewed the potential characteristics of ultrasonic motors from the friction control mechanism and divided into two categories, including the temporally controlled friction force and the spatially regulated friction force [5]. Littmann et al. proposed a model that is capable of predicting the reduction of the macroscopic friction force as a function of the ultrasonic vibration frequency, amplitude, and the macroscopic-sliding velocity [53]. Additionally, Dong et al. found that the operational noise was related to frictional materials, preload force, the contact area and interface condition, and the operational frequency [54]. Zhang et al. found that surface roughness is reduced to varying degrees in the metallic sliders due to ultrasonic polishing and/or micro-rolling effect [29]. For many structure types of standing wave linear ultrasonic motors [55,56], the stators have the characteristic of intermittent contact [57,58], and the micro-driving behavior is not unified and needs further investigation for friction compensation control and high-precision modulation [21,59].

The contact friction model based on Coulomb friction holds that the stator has transient viscous behavior only at the isokinetic point, and the tangential contact force has an abrupt change in the contact process of the stator [39,60,61]. For ultrasonic motors, the friction drive described by Hertzian contact and Coulomb friction theories is highly nonlinear [62]. As in the general friction cases, the switching of sticking and slipping is critical for ultrasonic motors. Changes in the friction force of sliding motion would affect the asperities that are subjected to excited tangential contact vibration; the Dahl model is valid to express such issues with tangential contact deformability, including the phenomenon of “pre-sliding displacement” [63]. Li et al. established a dynamic model using the energy method [64,65] and emphasized that the stick–slip phenomenon is easy to occur when the preload or the amplitude of stator vibration was large. Zhang et al. established a dynamic contact model based on the elastic contact theory and modified the Coulomb friction model considering tangential stiffness and preparatory displacement [39,57]. Additionally, Lv et al. proposed an analytical model for calculating the tangential force of the motor [66,67], which fully considers the influence of contact surface roughness. Their experimental data show that the model was more accurate than the equivalent spring model. The aforementioned researches are mostly based on Coulomb friction; their fundamental mechanical characteristics do not fully consider the dynamic behavior in the horizontal direction. Further, the static Coulomb model does not consider compliance of the contact zone and the vibration and sliding inertia of the friction body [68]. The Dahl friction theory [69] was proposed to consider the tangential friction force as a function of displacement. Li et al. proposed a full physical model of the standing-wave ultrasonic motor using longitudinal and bending vibration modes [70]. Their work considered the elastoplastic deformation characteristics and Dahl friction between the stator and slider [70], which is positive for friction modeling of linear ultrasonic motors. For ultrasonic motors, the plastic deformation of the interface materials or asperities is related to the cumulative effect of contact stress, which may be a key problem in terms of wear failure process rather than the micro friction drive modeling. Given the deficiency of the Coulomb friction model in describing the contact process of the stator/slider, a friction drive model of a V-shape linear ultrasonic motor based on Dahl friction theory will be established in this study. Importantly, we will analyze the piecewise and process characteristics of friction drive based on the impulse theorem and the energy transformation principle. The tangential compliance of the contact of the stator/slider is fully considered during the modelling of friction behavior for ultrasonic motors.

## 2. Friction Modelling of the Stator/Slider

### 2.1. Elliptic Motion at the End of Stator Tip

The V-shape vibrator has a left–right symmetrical structure. For simplification, the V-shape vibrator can be regarded as composed of two symmetrical Langevin transducers, which mainly vibrate in the axial direction. Then, the motion of the contact end of the V-shape vibrator can be given by the synthetic motion of the point. The displacement of the end of a single Langevin oscillator can be determined by the following formula.
(1)$$S=Asin\left(\omega t+\varphi^{*}\right)$$
where *A* is the amplitude of the end of the Langevin transducers in its axial direction, *w* is the angular frequency of the vibration, and *φ*^*^ is the initial phase of the vibration of the Langevin vibrators. If the sinusoidal excitation signal is connected to the right side of the oscillator and the cosine signal is connected to the left side, and the excitation mode is shown in Figure 3, the dis of the contact end of the V-shape vibrator can be expressed by Equation (2) in the *η*-*ξ* coordinate system.
(2)$$\{\begin{array}{c} u_{\eta }=usin\left(\omega t\right)\\ u_{\xi }=vcos\left(\omega t\right) \end{array}$$
where *u* and *v* are the amplitudes of the *η* direction and *ξ* direction of the vibrator tip in the *η*-*ξ* coordinate system. Then, the displacement of the end of the vibrator tip in the *x*-*y* coordinate system can be expressed by Equation (3).
(3)$$\{\begin{array}{c} u_{x}=u_{\eta }cos\theta -u_{\xi }cos\theta \\ u_{y}=u_{\eta }sin\theta +u_{\xi }sin\theta \end{array}$$

Formula (3) is an elliptic trajectory equation. *θ* is the angle between the *x*-axis and *η*-axis, which is generally 45°.

### 2.2. Dynamic Normal Force Analysis

Before the contact analysis, the following assumptions can be proposed: (1) the displacement of the stator mass center in the vertical direction can be ignored when the motor works in a steady state, and the stator mass center in the vertical direction is definite; (2) the vibration characteristics of the stator are not affected by the contact process. Taking the slider as the research object, the force diagram of the actor is shown in (a) and (b) of Figure 4. From the impulse theorem, it can be obtained that the sum of the impulse in the vertical direction of the stator in a contact period of the stator is equal to 0, then
(4)$$\int_{t_{1}}^{t_{2}}F_{N}\left(t\right)dt-\left(F_{0}+mg\right)T=0$$
where *t*_1_ is the time from the separation stage to the contact stage, *t*_2_ is the time from the contact stage to the separation stage, *T* is the theoretical full contact period of the stator and slider, *F*_0_ is the initial preload, and *m* is the effective mass of the stator. In this paper, the stator tip is cylindrical, and *F_N_* can be obtained from Hertz contact.
(5)$$F_{N}=\frac{\pi }{4}E^{*}L\delta$$
where *E*^*^ is the reduced elastic modulus of the materials of stator and slider, *L* is the length of the cylindrical tip, and *δ* is the depth at which the stator tip is pressed into the slider, which can be obtained from Figure 4c.
(6)$$\delta =-u_{y}\left(t_{1}\right)-h_{0}$$

Then *F_N_* can be expressed by the following expression
(7)$$F_{N}\left(t\right)=\frac{\pi }{4}E^{*}L\left(-u_{y}\left(t\right)-h_{0}\right)$$

In the separation phase, the stator is not in contact with the slider, and the normal contact force between them is zero. Therefore, the normal contact force between the stator and slider in a contact period is
(8)$$F_{N}\left(t\right)=\{\begin{array}{c} 0\;\;\;\;\;\;\;\;\;\;\;\;\;\;\;\;\;\;\;\;\;\;\;\;\;\;\;\;\;\;\;\;\;\;\;\;\;\;\;\;\;\;-u_{y}\left(t\right)<h_{0}\;\\ \frac{\pi }{4}E^{*}L\left(-u_{y}\left(t\right)-h_{0}\right)\;\;\;\;\;\;-u_{y}\left(t\right)>h_{0} \end{array}$$

### 2.3. Dynamic Tangential Force Analysis

It is assumed that the running speed of the slider is constant, and the tangential velocity of the stator tip is the first derivative of the tangential displacement of the contact tip concerning time, that is
(9)$$v_{t}=\left(\dot{u_{x}}\right)=\omega cos\theta \left(ucos\left(\omega t\right)+vsin\left(\omega t\right)\right)$$

Further, the tangential velocity of the end of the stator tip at the beginning of contact with the slider is
(10)$$v_{t}\left(t_{1}\right)=\omega cos\theta \left(ucos\left(\omega t_{1}\right)+vsin\left(\omega t_{1}\right)\right)$$

The tangential velocity at the beginning of separation is
(11)$$v_{t}\left(t_{2}\right)=\omega cos\theta \left(ucos\left(\omega t_{2}\right)+vsin\left(\omega t_{2}\right)\right)$$

Let the time of *t*_3_ and *t*_4_ be the isokinetic point [62] (that is, the time when the tangential velocity of the end of the stator tip is equal to the velocity of the slider), then
(12)$$v_{t}\left(t_{3}\right)=v_{t}\left(t_{4}\right)=v_{s}$$

Note that $t_{1}<t_{3}<t_{4}<t_{2}$, there is a speed comprepaison relation that is $\left|v_{t}\right|<\left|v_{s}\right|$ during the phase of $t_{1}\to t_{3}$,$t_{4}\to t_{2}$, and $\left|v_{t}\right|>\left|v_{s}\right|$ at the phase of $t_{3}\to t_{4}$. *v*_s_ is the stable speed of the slider.

According to the analysis of the tangential velocity of the tip end of the stator during the contact period, we can carry out the following inferential analysis. If only Coulomb friction is considered, the tip hinders the motion of the slider, and the friction force does negative work during the phases of $t_{1}\to t_{3}$. The contact hinders the motion, and the friction does negative work, and the contact drives the motion of the slider, and the friction does positive work during the phase of $t_{3}\to t_{4}$. Factually, with the Coulomb model it is difficult to comprehensively describe the stick–slip motion between the stator and slider. The contact phase of the stator and slider has a displacement difference in the tangential direction. Because the vibration of the stator tip is high frequency and micro-amplitude, the direct effect of displacement difference is small. Thus, the tangential compliance of the contact interface should be fully considered in the tangential contact of the stator/slider.

The contact surface of the stator is not an ideal smooth surface with a certain roughness. Following the dynamic friction model described by Dahl, the micro-concavity of the contact surface is modelled by the micro-spring. When the tangential load is applied, the micro-spring will deflect along the direction of the friction resistance, and there is no real sliding friction. When the tangential force is large enough, the contact is broken, and sliding occurs immediately [63]. According to the Dahl friction model, the tangential displacement between the contact surfaces of the stator and slider can be composed of two components: elastic displacement, *z,* and sliding displacement, *w*, as shown in Figure 5.
(13)x=z+w

When there is no sliding between the stator, the friction of the contact interface can be expressed as
(14)$$F_{\tau }=k_{t}z$$
where *k_t_* is the tangential contact stiffness between the stators, which can be obtained according to the analytical method given by Mindlin [71]. *F_τ_* is less than or equal to the sliding friction force, *F_c_*, between the contact interfaces.
(15)$$F_{c}=\mu k_{n}\delta$$
where *μ* is the sliding friction coefficient between the stator and slider, *δ* is the indented depth of the slider by the stator tip, and *k_n_* is the normal contact stiffness between the stators, which can be obtained from Formula (5).
(16)$$k_{n}=\frac{\pi }{4}E^{*}L$$
where *E*^*^ is the reduced elastic modulus of the contact couple of the stator and slider. It depends on the elastic modulus (*E*_1_, *E*_2_) and Poisson’s ratio (*v*_1_, *v*_2_) of both friction samples, which can be calculated from the expression, $1/E^{*}=\left(1-v_{1}^{2}\right)/E_{1}+\left(1-v_{2}^{2}\right)/E_{2}$. Then, the maximum non-sliding displacement that can be provided by the stator tip at any time during the contact period is
(17)$$z_{max}=\frac{F_{c}}{k_{t}}\;\;(t_{1}<t<t_{2})$$

Based on the above analysis, it can be concluded that the micro-spring established by the Dahl friction model will first deflect before the sliding of the stator tip, and the sliding friction will occur only when the deflection reaches the maximum value. For the contact of the V-shaped vibrator tip and slider, it is the viscous stage before the occurrence of sliding friction.

During the contact stage of the stator and slider, the normal contact force is a function of time [29], and *z_max_* is also a function of time in the contact stage of a driving cycle. The tangential direction of the end of the stator is consistent with the motion direction of the actuator, but the velocity is different. There is a displacement difference in the tangential direction during the contact period of the stator, and the difference is
(18)$$∆x=v_{s}\left(t-t_{1}\right)-\int_{t_{1}}^{t}v_{t}dt\;\;\;\;\;(t_{1}<t<t_{2})$$

By comparing the size of *z_max_* and |∆x|, we can judge when the stator is sticking and when it is slipping friction during the contact period.

According to Formula (3) and Formula (4), after *u* and *v* are determined, then *t*_1_ and *t*_2_ can be determined. The tangential velocities of the stator tip at *t*_1_ and *t*_2_ can be obtained by Equations (10) and (11). By giving the value of *z_max_* in advance (0 < *v_s_* < *v_0_*), the graph of *z_max_* and |*x*| with time can be drawn, where *v*_0_ is the no-load speed of the motor. Three cases can be illustrated in Figure 6. *z_max_* and |*x*| in the case of $v_{s}<v_{t}\left(t_{1}\right)$ is shown in Figure 6a, and *z_max_* and |*x*| in the case of $v_{t}\left(t_{1}\right)<v_{s}<v_{0}$ is shown in Figure 6b, and z_max_ and |*x*| in the case of $v_{s}=v_{0}$ is shown in Figure 4c. *t** in Figure 6a,b is obtained given the condition of $\left|∆x\left(t^{*}\right)\right|=z\left(t^{*}\right)$. When $\left|∆x\right|<z_{max}$, the micro-spring established according to the Dahl friction model is deflected, and there is no sliding friction between the stator and slider; otherwise, there is sliding friction between the stator and slider of the linear ultrasonic motor.

Figure 7 is taken as the research object for analysis and shows the situation of the motor when there is no load. A contact stage is divided into several sections. As seen, $\left|∆x\right|<z_{max}\;$ in the phase of $t_{1}\to t_{6}$, and there is always $\left|∆x\right|>z_{max}$ in the phase of $\;t_{6}\to t_{2}$. Further, two typical phases of the deflections of micro spring are shown with consideration for the relative location and direction.

In a driving cycle, the sum of the work done by the friction between the stator and the slider and the work done by the friction between the slider and the guideway should be 0. The magnitude and positive/negative function of the friction work between the stator and slider is analyzed according to the sub-section of Figure 7.

During this period of $t_{1}\to t_{3}$, the velocity of the slider is greater than the tangential velocity of the contact, and therein is $\left|∆x\right|<z_{max}$. During this stage, deflection occurs in the micro-spring established by Dahl friction; the direction of deflection is the same as the motion of the slider. At this time, the direction of the tangential force is opposite to the direction of the motion of the slider. With the torque of the tangential force, $F_{\tau }=-k_{t}∆x$, the deflection in the Dahl model arises. Using the differential method, in any period of t+dt, the distance is $dx=\left(v_{s}-v_{t}\right)dt$ of the tangential force, and the tangential force does negative work. The tangential force and the acting distance are both functions of time; using the differential element method, the element work in d*t* time is expressed as
(19)$$dw=F_{\tau }dx$$

If both sides of the equation are integrated at the same time, the work done by the tangential force between the stator and slider in the phase of $t_{1}\to t_{3}$ is
(20)$$W_{1}=\int_{t_{1}}^{t_{3}}-k_{t}∆x\left(v_{s}-v_{t}\right)dt$$

The relation $\left|∆x\right|<z_{max}$ always is true in the phases of $t_{3}\to t_{5}$, $t_{5}\to t_{4},\;t_{4}\to t_{6}$. The work done by the tangential force in each period is shown as follows
(21)$$W_{2}=\int_{t_{3}}^{t_{5}}-k_{t}∆x\left(v_{s}-v_{t}\right)dt$$
(22)$$W_{3}=\int_{t_{5}}^{t_{4}}-k_{t}∆x\left(v_{s}-v_{t}\right)dt$$
(23)$$W_{4}=\int_{t_{4}}^{t_{6}}-k_{t}∆x\left(v_{s}-v_{t}\right)dt$$

The deflection is $z=z_{max}$ of the micro-spring during the phase of $t_{6}\to t_{2}$. At this stage, the normal contact force decreases continuously, and as a result, the deflection of the micro-spring is not enough to offset the change of the displacement difference between the stator/slider in the tangential direction. At this point, the micro-spring will automatically return to the maximum value, *z*_max_, that can be provided at a certain time, and then the deflection of the micro-spring will gradually return to 0. Due to the self-recovery of the micro-spring, there will be sliding friction. The deflection direction of the micro-spring is opposite of the slider. Due to the recovery of the micro-spring, the tangential force does positive work, and the tangential force is $F_{C}=\mu k_{n}\delta$. In addition, the velocity of the slider is greater than the tangential speed of the stator tip during this phase, and the acting distance of the tangential force in any period of t+dt is shown as
(24)$$dl=d\left(\frac{F_{C}}{k_{t}}-\left|∆x\left(t_{6}\right)-∆x\left(t\right)\right|\right)$$

The work by friction force is expressed as
(25)$$W_{5}=\int_{t_{6}}^{t_{2}}F_{C}\frac{dl}{dt}dt)$$

The above content focuses on the work done by the driving force in different phases during the contact of the stator/slider. Here, the work done by the friction between the stator and the slide is further analyzed.

When the stator and the slider are in the non-contact stage, the negative work done by the friction between the slider and the rail is
(26)$$W_{6}=-m_{d}g\mu_{d}v_{s}\left(T-t_{2}+t_{1}\right)$$
where *m_d_* is the mass of the slider, and *μ_d_* is the coefficient of friction between the slider and the rail. When the stator and the slider are in contact, the negative work done by the friction between the slider and the rail is expressed as
(27)$$W_{7}=-v_{s}\left(t_{2}-t_{1}\right)\times \mu_{d}\left(\left(\frac{1}{t_{2}-t_{1}}\right)\int_{t_{1}}^{t_{2}}\left(m_{d}g+k_{n}\delta \right)dt\right)$$

As aforementioned, the sum of the work done by the tangential force of the stator/slider and the friction force between the slider and the rail is zero, then
(28)$$W_{1}+W_{2}+W_{3}+W_{4}+W_{5}+W_{6}+W_{7}=0$$

With consideration for the isokinetic point (*t_3_, t_4_*) of the tangential velocity of the stator tip, then
(29)$$v_{t}(t_{3})=v_{t}\left(t_{4}\right)$$

The deflection of the micro-spring is 0 at *t*_5_, then
(30)$$v_{s}\left(t_{5}-t_{1}\right)-\left(u_{x}\left(t_{5}\right)-u_{x}\left(t_{1}\right)\right)=0$$

The deflection of the micro-spring at time *t*_6_ is equal to $z_{max}\left(t_{6}\right)$, then
(31)$$\mu k_{n}\delta /k_{t}=∆x\left(t_{6}\right)\;$$

Solving expressions (29)–(32) simultaneously, *t*_3_, *t*_4_, *t*_5_, and *t*_6_ can be obtained. Further, the tangential force between the stator and slider in a driving period can be obtained, that is,
(32)$$F_{\tau }\left(t\right)=\{\begin{array}{c} -k_{t}∆x\left(t\right)\;\;\;\;\;\;\;\;\;t\in \left(t_{1},t_{6}\right)\\ \mu k_{n}\delta \;\;\;\;\;\;\;\;\;\;\;\;\;\;\;\;t\in \left(t_{6},t_{2}\right)\; \end{array}$$

At the same time, the running speed of the slider can also be obtained. Based on the above analysis, the tangential force between the stator and slider is periodic. Therefore, the output thrust of the motor during the steady-state operation should be equal to the force of the slider in a driving period. The time average value of the resultant force parallel to the running direction is
(33)$$F_{p}=\frac{1}{T}\int_{0}^{T}c_{a}\left(F_{\tau }\left(t\right)-f\right)dt$$
where *c_a_* is the nominal contact area ratio with consideration for the bearing length ratio ($∆M_{r}=M_{r2}-M_{r1}$), *f* is the friction force between the slider and the rail. The nominal contact area ratio can be expressed by the following equation in two dimensions:(34)$$\{\begin{array}{c} c_{a}=1-{\left(1-∆M_{r}\right)}^{2}\\ ∆M_{r}=\varphi \left(E,\vartheta ,F_{n,\;}R_{a},F_{0}\right)\;\; \end{array}$$
where the bearing length ratio, $∆M_{r}$ can be seen as a function of the material mechanical properties (e.g., elastic modulus, *E*, Posson’s ratio, *ϑ*), surface properties of the contact couple (surface roughness), and the applied preload *F*_0_ and normal load *F_n_*. 

The above analysis is conducted after Figure 7, and the contact process mainly focuses on the friction couple of the stator/slider. Specifically, Figure 7 shows the situation when the ultrasonic motor is no-load. In the cases shown in Figure 6a,b, the motor is running with a load. The slider speed with loading is lower than the speed of the motor without loading. Combined with Figure 6a, it can be seen that since the speed of the slider is always smaller than the tangential speed of the contact during the contact period of the stator and the slider, the tangential contact force between the stator and the slider will always do positive work. In Figure 6b, $v_{t}\left(t_{1}\right)<v_{s}<v_{0}$, and the tangential contact force will still do negative work for a period of time when the stator and slider first contact and then start to do positive work. Similarly, the same method can be used to analyze any of the situations for linear ultrasonic motors by establishing the equation(s) and solving after substituting the parameters.

## 3. Driving Characteristic Simulation

### 3.1. Simulation Flowchart and Parameters

According to the aforementioned model, the influence of parameters on the performance of the V-shaped linear ultrasonic motor was studied by setting relevant parameters. The output performance of the ultrasonic motor was predicted by numerical simulation, and the performance of the motor was evaluated. The simulation results were compared with the drive model based on the Coulomb friction. The relevant parameters were set following Table 1 and Table 2. Following the aforementioned theoretical analysis of linear ultrasonic motors, a flowchart of theoretical calculation for the mechanical performances of linear ultrasonic motors is provided in Figure 8. 

### 3.2. Tangential Contact Force

In this study, three aspects of influence factors covering preload, vibration amplitude, and stiffness were considered. Under different preloads, the variation of the tangential force between the stators and sliders in a driving cycle is shown in Figure 9(ai). When the amplitude is constant, the contact time of the stator becomes longer when the pre-load increases, and the larger the preload is, the larger the variation range of *F*_τ_ is. The increasing proportion of tangential force doing negative work is larger than that of doing positive work; thus, the increase of pre-pressure reduces the proportion of positive work done by tangential contact force, as shown in Figure 9(bi), which is also the reason why the no-load speed of the motor decreases with the pre-pressure. Figure 9(aii) shows the tangential contact force between the stators in a driving period excited by different amplitudes of the oscillator. With the increase of the amplitude of the oscillator, the tangential contact force of the stator driving the negative work becomes smaller. The tangential contact force of doing positive work becomes larger, and it can be determined that the proportion of the tangential contact force doing positive work increases, as shown in Figure 9(bii). Thus, the speed of the motor will increase. It can be seen from Figure 9(aiii) that the lower the contact stiffness of the stator is, the longer the contact time of the stator is, and the greater the tangential force of the negative work done by the stator driving the stator is, the smaller the tangential force of doing positive work is. Therefore, under the same driving conditions, the higher the contact stiffness of the stator, the higher the proportion of positive work done by the tangential force, as shown in Figure 9(biii). The selection of stator materials with higher elastic modulus is helpful to improve the output performance of the motor.

During a friction-driving period of the stator/slider, the tangential contact force between the stator is shown in Figure 10. It can be seen from the diagram that there is a significant difference in the tangential force calculated by the Coulomb model and the Dahl model at any time in the contact stage. At the same time, there is a sudden change in the tangential contact force between the stator based on the Coulomb model. In the driving period of the stator, the contact process of the stator is continuous. The continuous change of the contact state does not tend to cause the abrupt change of the tangential contact force, but the tangential contact force between the stators based on the Dahl model is continuous. Compared with the Coulomb friction model, the friction drive model based on the Dahl friction theory is more consistent with the actual contact condition of the stator and slider.

### 3.3. Asymmetry of Forwarding and Reverse Speeds

Due to the machining errors and assembly errors of each part of the stator, the performances of the left and right sides of the V-shaped vibrator are not the same. The most intuitive difference is reflected in the amplitudes on both sides of the vibrator. The amplitudes on the left and right sides of the vibrator are not exactly equal, and the motion trajectory of the contact end will be an oblique ellipse. The amplitude value on the right side of the fixed V-shape vibrator remains unchanged, and the amplitude value on the other side is changed, such as *u* = 1 μm, and *v* changes in the range of 0.9–1 μm. The obtained tangential contact force between the stator and the slider is shown in Figure 11a, and the left and right running speeds of the slider are shown in Figure 11b. It can be seen from the figure that with the increase of the amplitude difference between the left and right sides of the V-shape vibrator, the left and right running speeds of the slider decrease, and the speed of the slider running to the right decreases faster, and the speed difference between the left and right running is larger. This shows that when the amplitude on the right side of the V-shape vibrator is greater than the amplitude on the left side, more positive work is done by the tangential contact force when the slider moves to the left than when the slider moves to the right. However, the driving model established by Coulomb friction cannot simulate the speed difference between the left and right running of the motor, which is inconsistent with reality. Therefore, the friction-driving model based on the Dahl friction is better than the Coulomb friction model for V-shape linear ultrasonic motors.

## 4. Experimental Detail and Results

### 4.1. Experimental Apparatus and Test Method

The self-made V-shape linear ultrasonic motor and its performance test method are used in the experiment, as shown in Figure 12. The V-shape vibrator is assembled in an aluminum shell, which is connected to the base using the four groups of springs. A movable slider is set on the top surface of the sliding block, which is connected to the I-beam, and the linear unit is fixed on the base. When a tension spring is fixed on the slider, the spring will be elongated when the slider moves in a straight line. At the same time, the displacement data of the vibrator is collected by a macroscopic laser displacement sensor. The thrust force and the output speed of the ultrasonic motor at a certain time are obtained through data processing. The pulling force received by the actuator is the output thrust of the motor, and the tension spring can only be removed when measuring the no-load speed of the motor. The detail of the measurement apparatus and approach can be referred to in the literature [22].

### 4.2. Experimental Verification

The comparison between the typical working condition calculation results and the test results of the two models is shown in Figure 13. As can be seen, compared with the Coulomb model, the simulation results based on the Dahl model are closer to the experimental results, and the trend of the simulation is consistent with the experimental results. When the pre-pressure and excitation voltage are small, the simulation results are very close to the experimental results. Because the effects of inertia force and large preload on the resonant frequency of the oscillator are not taken into account in the model, the simulation results begin to differ from the experimental results when the pre-pressure and excitation voltage are large. As seen, the change of the peak driving voltage has no significant effect on the accuracy of the simulated friction models. Differently, the preload has a clear influence on the output speed, and there is a significant difference between the simulation results and the experimental data. 

For this reason, taking the preload as the external control condition, the relative errors of the two friction drive models are compared and analyzed, as shown in Figure 14. It can be seen from the figure that the absolute values of the relative errors of the two friction models increase with the increase of the preload. However, under the normal working conditions of the ultrasonic motor, the relative error of the Dahl friction model is less than 5%, which is of more reference significance in engineering. The threshold of relative error also directly proves that the Dahl friction model has good adaptability and application.

The comparison between the theoretical and experimental mechanical characteristic curves is shown in Figure 15. As seen, the simulation results of mechanical characteristics are consistent with the experimental results, and the two results are close when the load is small, indicating that the Dahl model can describe the driving process of the motor and can predict the performance of the motor. At the same time, it is also noted that when the load is large, the simulation results begin to deviate from the experimental results, which may be because the velocity of the slider is regarded as constant when modelling. Factually, the velocity of the slider varies periodically and microscopically. When the load is large, the motion state of the slider is greatly affected by the load, and there will be a deviation between the simulation and experimental results.

## 5. Conclusions

Based on the Dahl friction theory, a friction-driving model of linear ultrasonic motor was established, and the tangential compliance between the contact interfaces of the stator was considered. In addition, the changing trend of tangential force in different periods in the contact stage was analyzed. The friction-driving process of a V-shape linear ultrasonic motor was analyzed based on the Dahl friction model, and the force transmission process between the stator and slider is directly reflected by numerical simulation. This study is capable of providing a certain reference for understanding the micro-driving mechanism of the linear standing-wave ultrasonic motors/actuators. Some main conclusions are obtained as follows. 

(1) The numerical simulation results show that the increase of preload increases the proportion of negative work done by tangential contact force, and the increase of contact stiffness of the stator increases the proportion of positive work done by tangential contact force. Compared with the simulation results of the Coulomb friction model, the model established in this study is continuous and conforms to the physical rules. Further, the relative error threshold is less than 5%. (2) The difference between the left and right running speed accounted for the inconsistent performance of the left and right sides of the V-shaped oscillator, which leads to a different proportion of positive work done by the tangential contact force when the motor is running left and right and which leads to the difference in the left and right running speed of the ultrasonic motor. (3) Compared with the Coulomb friction model, the established Dahl friction model in this study is capable of continually calculating the tangential force and horizontal dynamics in accordance with the physical laws. 

## Figures and Tables

**Figure 1 micromachines-13-01407-f001:**
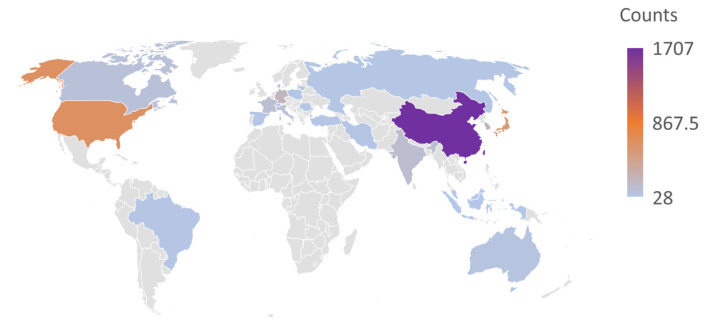
Published map of linear ultrasonic motors collected from Web of Science Core Collection^®^ date to 12 August 2022 (search item: ”linear ultrasonic motors” or ”linear piezoelectric motors”).

**Figure 2 micromachines-13-01407-f002:**
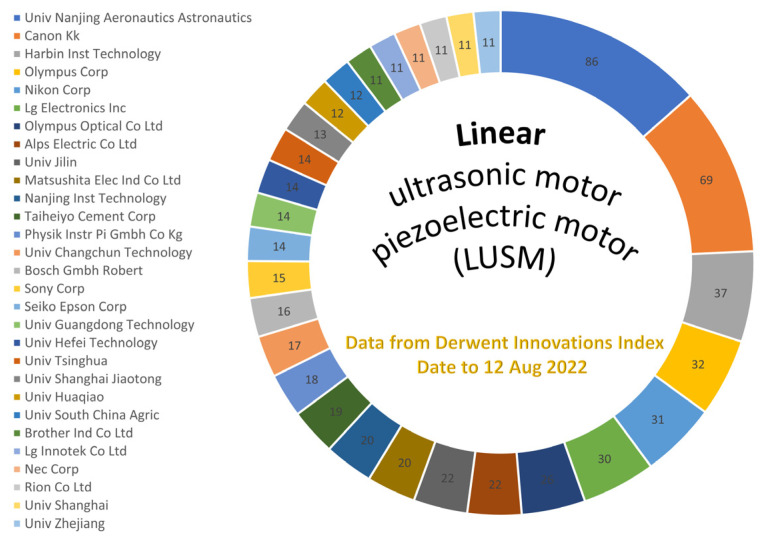
Patent counts of linear ultrasonic motors from Derwent Innovations Index^®^ date to 12 August 2022 (search item: “linear ultrasonic motors” or “linear piezoelectric motors”).

**Figure 3 micromachines-13-01407-f003:**
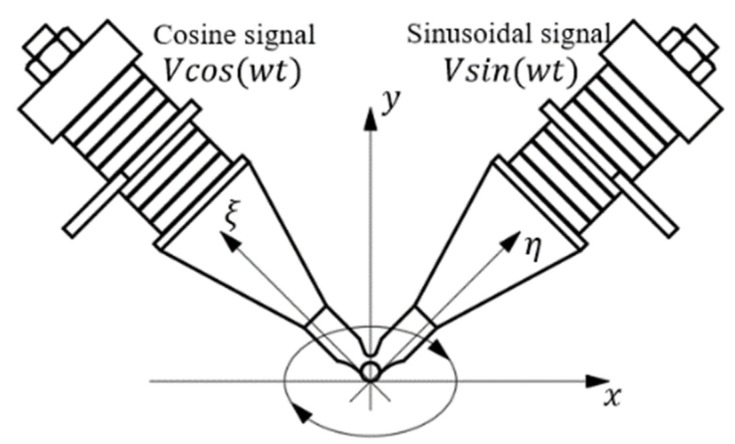
Excitation mode of the V-shape vibrator and ellipse track of the driving tip.

**Figure 4 micromachines-13-01407-f004:**
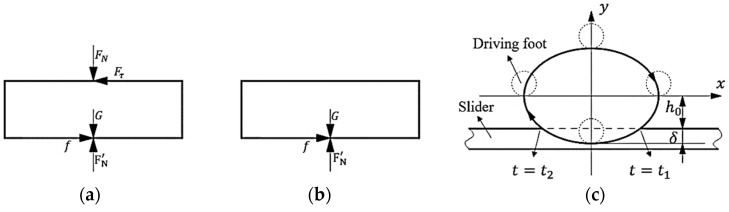
Static force diagram of the slider and the elliptical locus in one contact period. (**a**) Contact stage, (**b**) separation stage, and (**c**) elliptical locus in a contact period.

**Figure 5 micromachines-13-01407-f005:**
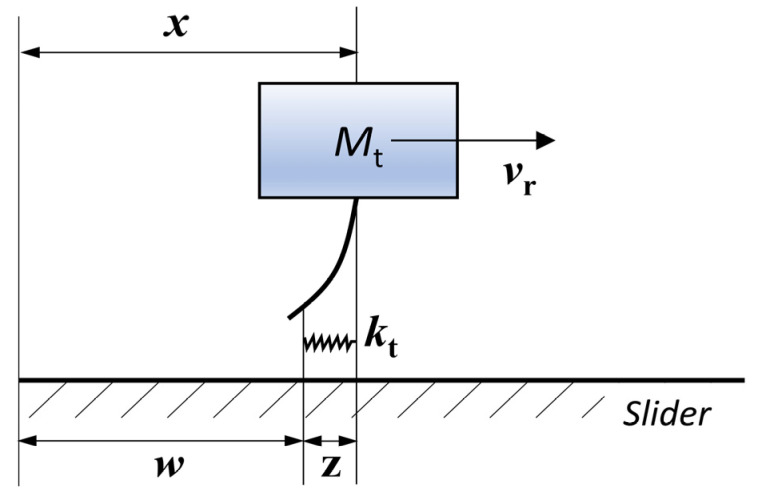
Illustration of Dahl friction model for the standing-wave ultrasonic motor.

**Figure 6 micromachines-13-01407-f006:**
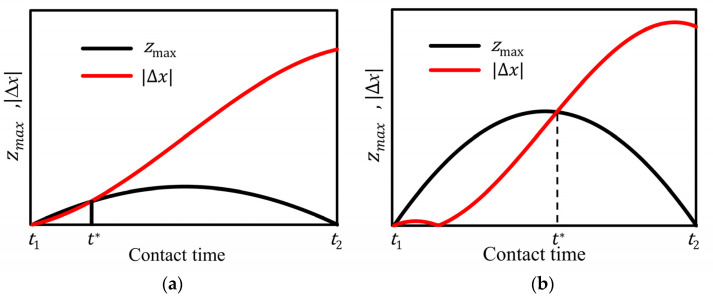
Curves of $z_{max}$
and |∆x| with respect to time corresponding to different $v_{s}$. (**a**) *v*_s_ < *v_t_*(*t*_1_), (**b**) *v_t_*(*t*_1_) < *v*_s_ < *v*_0_.

**Figure 7 micromachines-13-01407-f007:**
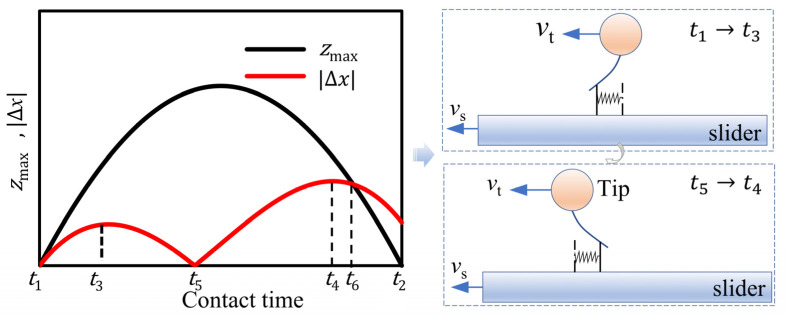
Deflection of the micro spring in a contact period and two typical phases.

**Figure 8 micromachines-13-01407-f008:**
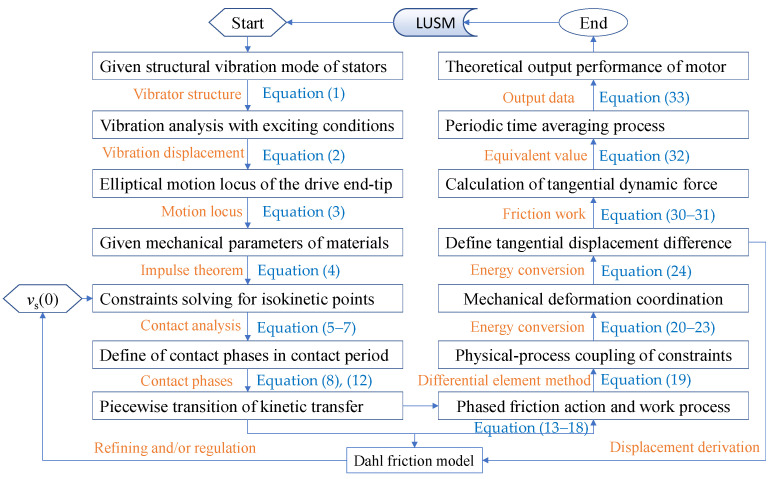
Calculating flowchart of Dahl friction model for ultrasonic motors in this study.

**Figure 9 micromachines-13-01407-f009:**
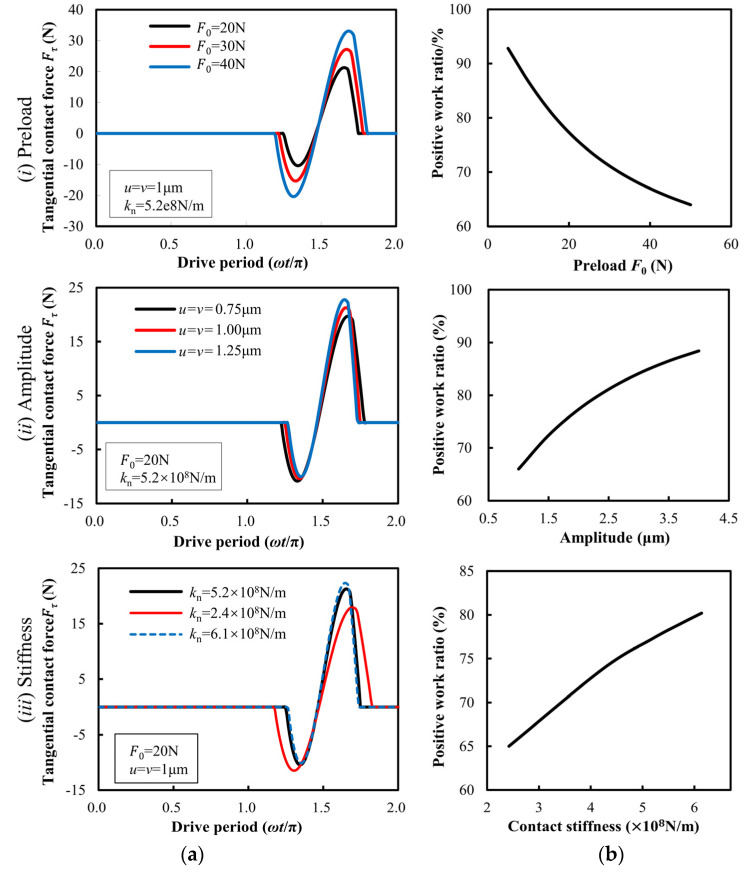
Tangential contact force between stator and slider and its proportion of positive work under different conditions. (**a**) Tangential contact force, and (**b**) positive work ratio.

**Figure 10 micromachines-13-01407-f010:**
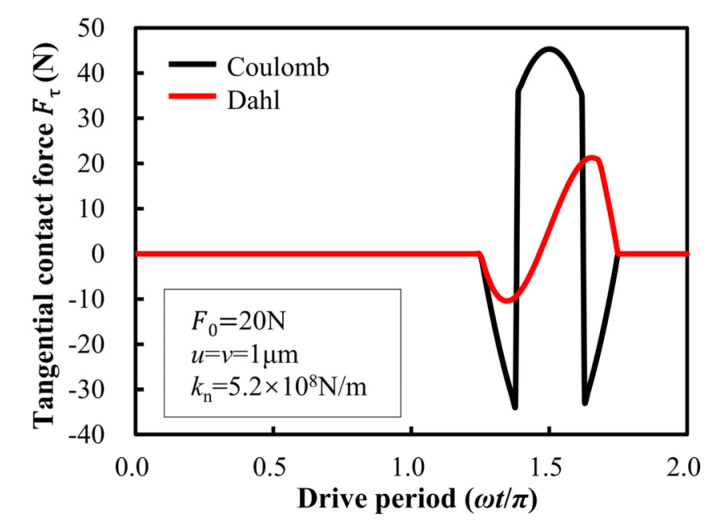
Periodic comparison of tangential contact forces for Dahl model and Coulomb model.

**Figure 11 micromachines-13-01407-f011:**
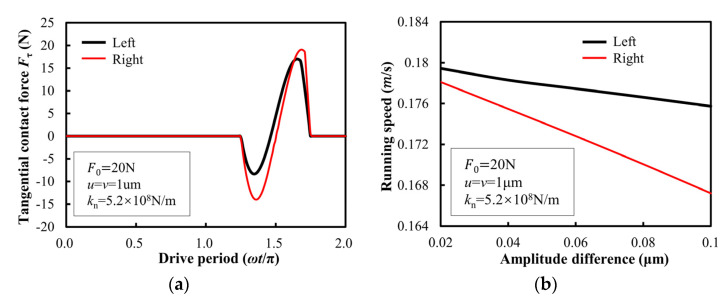
Asymmetry illustration of tangential contact force (**a**) and output speed (**b**) between stator and slider for the forward (e.g., right) and inverse (e.g., left) operations of the ultrasonic motor.

**Figure 12 micromachines-13-01407-f012:**
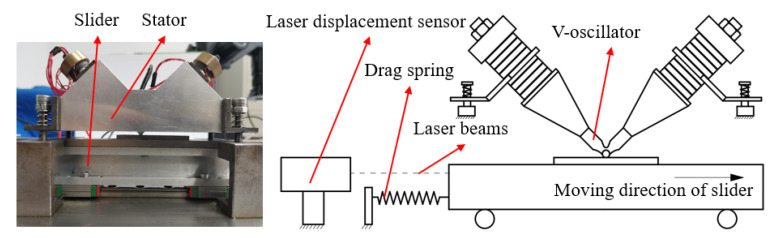
Illustration of a homemade ultrasonic motor and the testing sketch for friction-driving performances.

**Figure 13 micromachines-13-01407-f013:**
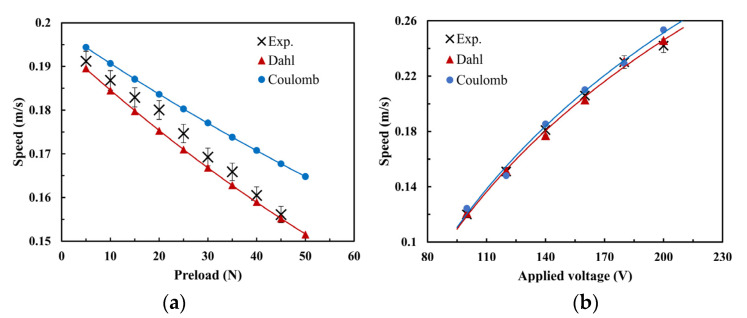
Comparison illustration of no-load theoretic and experimental speeds of the ultrasonic motor under different working conditions: (**a**) preload and (**b**) applied voltage.

**Figure 14 micromachines-13-01407-f014:**
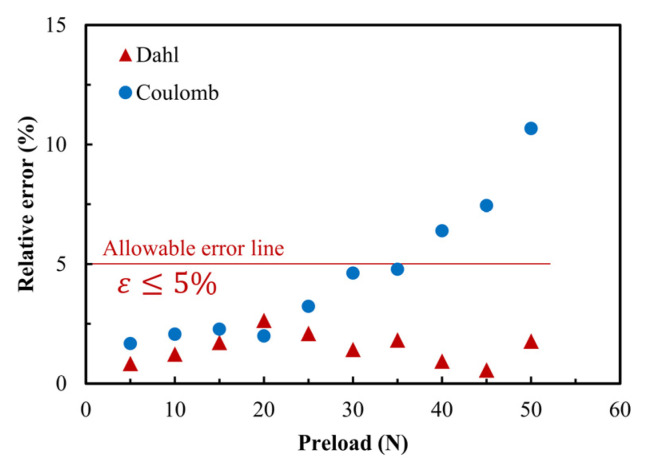
Comparative analysis of relative error of output velocity based on preload change for Dahl and Coulomb friction models. The bold black line reads the reference line, and the blue line reads the allowable error line in engineering sense.

**Figure 15 micromachines-13-01407-f015:**
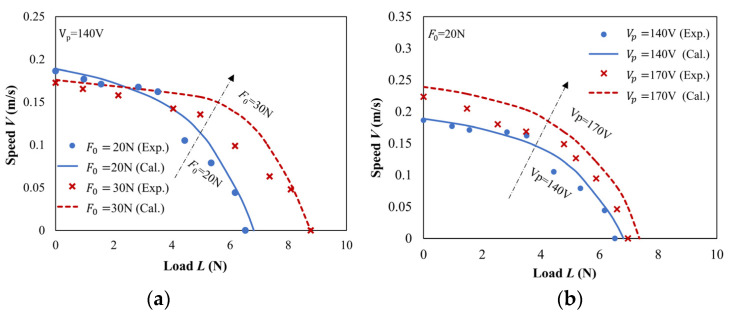
Comparison illustration of theoretical and experimental curves of mechanical characteristics for ultrasonic motors under different working conditions: (**a**) preload and (**b**) applied voltage.

**Table 1 micromachines-13-01407-t001:** Detail illustration of the material parameters of vibrator parts.

Part	Material	Density (g/cm^3^)	Elastic Modulus (GPa)	Poison’s Ratio
End-cover	Brass	8.73	108	0.31
Electrode	Copper	8.96	110	0.32
Front-cover	2A12	2.70	70	0.3
Bolt	304 steel	7.93	209	0.24
Nut	304 steel	7.93	209	0.24
Tip	Alumina	3.70	300	0.2
Piezo ring	PZT4	7.45	83.3	0.3

**Table 2 micromachines-13-01407-t002:** Simulation parameters for friction-driving process of the ultrasonic motor.

Parameter	Symbol	Value	Unit
Elastic modulus of drive tip	*E_1_*	300	GPa
Vibration amplitude	*A*	0.5~4	μm
Stator mass	*m_e_*	273	g
Slider mass	*m_s_*	273	g
Applied preload	*F* _0_	5~50	N
Applied voltage	*V_p_*	100~200	V
Normal stiffness	*k_n_*	2.0~6.5 × 10^7^	N/m
Tangential stiffness	*k_t_*	0.4~1.3 × 10^7^	N/m
Exciting frequency	*f*	32.4	kHz
Surface roughness	*R_a_*	0.25	μm
Contact ratio coefficient	*c_a_*	0.85~0.99	−
COF of tip and slider	*μ*	0.3	−
COF of slider assembly	*μ_d_*	0.004	−

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
