# Peer review of "Dahl Friction Model for Driving Characteristics of V-Shape Linear Ultrasonic Motors"

_micromachines, 2022, doi:10.3390/mi13091407_

Round 1
Reviewer 1 Report
In general, the paper is interesting. However a few major and minor remarks I have.
First of all, the introduction should be rewritten. There are a lot of old papers from 90', and it would help if you focused on new and recent trends. Moreover, the bibliography should include the EU and US papers, not only Asian. In my opinion, this is an unreliable approach to literature review.
The second and third chapters of the modeling and simulations are clear. However, the chapter on experimental analysis is not clear to me.
You should compare simulation and experimental results. In simulations, you presented force vs drive period, and in the experimetal tests speed vs preload. Why?
You stated: „These simulation results of output performances and mechanical characteristics of ultrasonic motors are consistent with the experimental results.”
However, I do not know where those results are?
Please, do more experimental tests and compare them with simulations. Or change the simulation results.
Author Response
Comments and Suggestions for Authors
In general, the paper is interesting. However, a few major and minor remarks I have.
Comment (1): First of all, the introduction should be rewritten. There are a lot of old papers from 90', and it would help if you focused on new and recent trends. Moreover, the bibliography should include the EU and US papers, not only Asian. In my opinion, this is an unreliable approach to literature review.
Response: Thank you for your comments and criticism. We have improved the introduction, with consideration of the related progress from the EU and USA.
(2) The second and third chapters of the modelling and simulations are clear. However, the chapter on experimental analysis is not clear to me.
Response: We have revised the experimental details in the revised manuscript.
(3) You should compare simulation and experimental results. In simulations, you presented force vs drive period, and in the experimental tests speed vs preload. Why?
Response: Thanks for your good comments. This study focuses on friction driving process of linear ultrasonic motors using an improved Dahl friction model with consideration of ultrasonic vibration. Theoretically, enough parameters can be obtained from the simulation data for given enough initial conditions. Differently, we can obtain the limited experimental parameters for the limited measuring conditions and apparatus. Specifically, the measurement of drive period is unavailable for us. In the experimental results, we provide the controllable parameters (e.g., preload, voltage) and the output performances (i.e., thrust force and speed) for the classical mechanical characteristics.
(4) You stated: “These simulation results of output performances and mechanical characteristics of ultrasonic motors are consistent with the experimental results.” However, I do not know where those results are.
Response: We have improved these unclarified expressions in the revised manuscript. Thanks.
(5) Please, do more experimental tests and compare them with simulations. Or change the simulation results.
Response: Thank you very much for your excellent comments. Your suggestions are good and important for the level improvement and data enrichment of our paper. Some issues and reasons have been illustrated in the question (3). Through our efforts, further improvement and progress both in theoretical and experimental would be provided in the following studies.

Reviewer 2 Report
The authors present a driving model of V-shaped linear ultrasonic motor based on Dahl friction model. The proposed model result shows better performance than Coulomb friction model. However, given the multiple published works on the same or similar modeling method, the proposed model lacks novelty.
To help understand readers I propose below:
1. What is the problem with the previous works? Similar modeling of an ultrasonic motor had already been studied in [1]. Although the stator of the motor in [1] is different from V-shaped stator, the motion trajectory of the driving foot of both motors are the same, they are all elliptical motions. Dahl friction model and stick-slip motion have been considered in [1].Therefore, please clarify the research gap addressed in your work?
[1] X. Li, Z. Chen, Z. Yao, Contact analysis and performance evaluation of standing wave linear ultrasonic motors via a physics-based contact model, Smart Mater. Struct. 28 (2019), 015032, https://doi.org/10.1088/1361-665X/aaf11a.
2. The expression for FN(t) in equation (7) and equation (8) are not consistent. The Operation symbol “+” in equation (8) should be “-“.
3. The symbol Fn in line 98 and line 107 should be FN.
4. Some variables are not explained in the manuscript, such as E*, T, vs.
5. In line 158, “between the stators” may be between the stator and the slider. I believe there is no sliding between stators.
6. What is the meaning of k2 in equation (28)? Should it be kn?
7. In section 3, ”tangential contact force” may be “tangential friction force”. It is somewhat misleading, because contact force is usually associated with normal contact force.
8. It is recommended to show how the model is solved with graph.
9. Please provide the parameters used in the simulation, such as geometric parameters and material parameters of the motor. These parameters can help readers perform similar analyses.
10. Page 9, line 313, “It shows that compared with the Coulomb friction model, the friction drive model based on Dahl friction theory is more consistent with the actual contact condition of the stator”. However, there is no actual contact condition shown in figure 6. I can't understand how to draw the conclusion that Dahl friction is more consistent with the actual contact condition.
11. Given the large errors between the measured and simulated results in figure 10, it is inappropriate to say “the simulation results of mechanical characteristics are consistent with the experimental results”. The reasons for the errors need to be further explained.
12. In conclusion, please briefly describe the results of the study without repeating the text body. In particular, advantages over similar studies should be identified. If you have any, prospects, please add them.
Author Response
Comments and Suggestions for Authors
The authors present a driving model of V-shaped linear ultrasonic motor based on Dahl friction model. The proposed model result shows better performance than Coulomb friction model. However, given the multiple published works on the same or similar modeling method, the proposed model lacks novelty. To help understand readers I propose below:
Comment 1. What is the problem with the previous works? Similar modelling of an ultrasonic motor had already been studied in [1]. Although the stator of the motor in [1] is different from the V-shaped stator, the motion trajectory of the driving foot of both motors is the same, they are all elliptical motions. Dahl friction model and stick-slip motion have been considered in [1]. Therefore, please clarify the research gap addressed in your work.
[1] X. Li, Z. Chen, Z. Yao, Contact analysis and performance evaluation of standing wave linear ultrasonic motors via a physics-based contact model, Smart Mater. Struct. 28 (2019), 015032, https://doi.org/10.1088/1361-665X/aaf11a.
Response: Thanks for your kind comments and suggestion. The abovementioned ref [1] has been supplementally cited in the revised manuscript. The difference between the work of this paper and the reference [1] lies in three aspects: (1) The given Ref [1] considers both elastic deformation and plastic deformation, and there is cognitive difference in this problem. Our authors believe that for the problem of friction drive between elastic stator and slider, the plastic deformation of the interface is related to the cumulative effect of contact stress, which may be a key problem in terms of wear failure process rather than the micro friction drive modeling. (2) From the point of view of systematic modeling, the type of electromechanical coupling model is necessary. However, the relevant parameters are based on the similarity theory of the components of electromechanical system, and it is difficult to provide direct physical reference for friction drive. On this issue, there is a difference between the given reference [1] and our paper. (3) In our paper, we analyze the piecewise and process characteristics of friction drive based on momentum theorem and energy transformation principle, which is very different from the given reference [1]. In addition, the reviewers also mentioned that the stator structure of the standing wave ultrasonic linear motor is also different. We hope these issues are enough supporting for academic independence of our paper. We thank the reviewers for their constructive comments and questions.
Comment 2. The expression for FN(t) in equation (7) and equation (8) is not consistent. The Operation symbol “+” in equation (8) should be “-“.
Response: Thanks for your kind comment. We have modified the error in the revised manuscript.
Comment 3. The symbol Fn in line 98 and line 107 should be FN.
Response: Thanks. We have corrected them in the revised manuscript.
Comment 4. Some variables are not explained in the manuscript, such as E*, T, vs.
Response: We have supplied the necessary explanation in the revised manuscript.
Comment 5. In line 158, “between the stators” may be between the stator and the slider. I believe there is no sliding between stators.
Response: We have revised this error. Thanks.
Comment 6. What is the meaning of k2 in equation (28)? Should it be kn?
Response: Thanks. We have corrected it in the revised manuscript.
Comment 7. In section 3, “tangential contact force” may be “tangential friction force”. It is somewhat misleading because contact force is usually associated with normal contact force.
Response: Tangential contact force is more general and does not contribute the friction force and friction work. If it is replaced by the “tangential friction force”, it would arise a mislead of effective friction power during the friction drive process. In addition, there is some literatures have used this expression during the dynamic friction process of ultrasonic motors. Thanks.
Comment 8. It is recommended to show how the model is solved with a graph.
Response: According to your comment, we have supplied a calculated flowchart for the given model in the revised manuscript.
Comment 9. Please provide the parameters used in the simulation, such as geometric parameters and material parameters of the motor. These parameters can help readers perform similar analyses.
Response: Thanks. We have supplied the simulated parameters in the manuscript, as shown in Fig. 5-7. The geometrical and material parameters have been supplied in the revised manuscript, as shown in the additional related citation.
Comment 10. Page 9, line 313, “It shows that compared with the Coulomb friction model, the friction drive model based on Dahl friction theory is more consistent with the actual contact condition of the stator”. However, there is no actual contact condition shown in figure 6. I can't understand how to draw the conclusion that Dahl friction is more consistent with the actual contact condition.
Response: Thank you for your concise comment. We compared the mechanical characteristics curves of ultrasonic motors from theoretical and experimental data. Factually, we have no actual contact condition and can’t draw that conclusion. We have deleted this sentence in the revised manuscript.
Comment 11. Given the large errors between the measured and simulated results in figure 10, it is inappropriate to say “the simulation results of mechanical characteristics are consistent with the experimental results”. The reasons for the errors need to be further explained.
Response: Thanks for your good comments. We have revised these expressions in the revised manuscript.
Comment 12. In conclusion, please briefly describe the results of the study without repeating the text body. In particular, advantages over similar studies should be identified. If you have any, prospects, please add them.
Response: Thanks. We have improved the conclusion section in the revised manuscript.

Round 2
Reviewer 1 Report
Unfortunately, the authors did some changes in the introduction, but this is not enough. As I said, it should be rewritten (from the beginning), not only adding one or two sentences. Some references are very old. It is not clear to me why you use such old literature.
The error in theoretical and experimental results is still significant. It is not applicable. Please, do the experiments with higher precision or calculate redefine the theory.
Author Response
Comment 1. Unfortunately, the authors did some changes in the introduction, but this is not enough. As I said, it should be rewritten (from the beginning), not only adding one or two sentences. Some references are very old. It is not clear to me why you use such old literature.
Response: Thanks for your comments. According to the review’s comment, we have revised the whole section of the Introduction, including the improvement of sentences and renewing of pieces of literature.
Comment 2. The error in theoretical and experimental results is still significant. It is not applicable. Please, do the experiments with higher precision or calculate and redefine the theory.
Response: Thanks for your good comment. After our careful checking, we find that we cannot consider the effect of the theoretical contact area in the previous simulation model, i.e., the asperities of space distribution for the contact area. During the simulation process, we considered the contact stiffness and the controllable contact boundary (i.e., contact width). Factually, for a real dynamic contact process, the contact area is not only dominated by the contact boundary on a 1D scale. Therefore, we further consider the nominal contact area ratio resulting from the horizontal roughness and bearing length ratio. An important and reasonable assumption is the basis of the isotropic material property of bristle stiffness on the two-dimensional scale of the effective contact interface. When coupling the bearing length ratio, the simulation errors were reduced clearly. The original simulated data showed that the effect of the preload is unsatisfactory. After refining the simulation model, the errors of speed are reduced clearly, as shown in Fig. 1S and 2S, referring to Figures. 13 and 14. Further, the simulated data of the mechanical characteristics of ultrasonic motors is consistent with that of experimental ones. Consequently, the quality of our manuscript has been significantly improved. We sincerely appreciate the encouragement of the editors and the rigorous demands of the reviewers.

Reviewer 2 Report
The authors have improved their manuscript. However, the novelty of the work is still not clearly presented. Here are some suggestions.
1. Although the authors have rewritten the introduction, the research gap between presented studies and existing studies is still not properly clarified.
2. There are some problems with the equations:
(1) The symbol w in Equations (1)(2)(9)(10)(11) should be ω (small Omega).
(2) Why there is an L in equation (17)? Should it be zmax=Fc/kt (t1<t<t2)?
(3) Is the parameter μd missing from equation (28)?
(4)There is a redundant 0 in Equation (32).
I recommend you check all the equations in the manuscript carefully.
3. The symbols in the equations should be defined. This is a basic requirement for research paper. There are still some symbols not defined in your manuscript, such as md, μd in equation (27). Please go through your manuscript and make sure all the symbols have been defined.
4. The authors claim that a calculated flowchart for the model is supplied in the revised manuscript. However, the flowchart is not found.
5. I still recommend providing the parameters of the prototype motor so the readers are able to perform similar analyses. Are the parameters shown in Fig. 5-7 sufficient to calculate the performance of the motor? Only four parameters, F0,u,v,and kn are shown in Fig. 5-7. I presume the parameters including but not limited to kt, m, μ, md, μd, ω are all necessary for calculation.
6. Same comment as mentioned in the previous manuscript. Given the large errors between the measured and simulated results in figure 10, it is inappropriate to say “the simulation results of mechanical characteristics are consistent with the experimental results”. Although the authors claim they have revised the expression, same expression still appears in the revised manuscript. The reasons for the errors need to be further explained. How the load affects the motion state of the slider?
Author Response
Comment 1. Although the authors have rewritten the introduction, the research gap between the presented studies and existing studies is still not properly clarified.
Response: We have improved the whole introduction in the revised manuscript. Thanks.
Comment 2. There are some problems with the equations:
(1) The symbol w in Equations (1)(2)(9)(10)(11) should be ω (small Omega).
(2) Why there is an L in equation (17)? Should it be zmax=Fc/kt (t1<t<t2)?
(3) Is the parameter μd missing from equation (28)?
(4) There is a redundant 0 in Equation (32).
I recommend you check all the equations in the manuscript carefully.
Response: Thanks for your good comments. We are sorry for the aforementioned errors in the expression and equations. We have checked all equations and moderated them in the revised manuscript.
Comment 3. The symbols in the equations should be defined. This is a basic requirement for a research paper. There are still some symbols not defined in your manuscript, such as md, μd in equation (27). Please go through your manuscript and make sure all the symbols have been defined.
Response: Thank you for your good comments. We have supplied the necessary symbols in the revised manuscript.
Comment 4. The authors claim that a calculated flowchart for the model is supplied in the revised manuscript. However, the flowchart is not found.
Response: We are sorry for the missing information in the last revision. We have added the flowchart to the revised manuscript.
Comment 5. I still recommend providing the parameters of the prototype motor so the readers are able to perform similar analyses. Are the parameters shown in Fig. 5-7 sufficient to calculate the performance of the motor? Only four parameters, F0,u,v,and kn are shown in Fig. 5-7. I presume the parameters including but not limited to kt, m, μ, md, μd, ω are all necessary for calculation.
Response: Thanks for your excellent comments. Four parameters are not enough to calculate the operation parameters for ultrasonic motors. We have supplied the others in the revised manuscript.
Comment 6. Same comment as mentioned in the previous manuscript. Given the large errors between the measured and simulated results in figure 10, it is inappropriate to say “the simulation results of mechanical characteristics are consistent with the experimental results”. Although the authors claim they have revised the expression, same expression still appears in the revised manuscript. The reasons for the errors need to be further explained. How the load affects the motion state of the slider?
Response: Thanks for your excellent comments. We have moderated the sentence “the simulation results of mechanical characteristics are consistent with the experimental results”. The simulation results of mechanical characteristics are not well consistent with the experimental results, because of the supposed conditions for simulation analysis and the measured errors for experimental values. Further, the errors arising from the preload are influenced by the original model, and we considered the 2D surface roughness and bearing length ratio during the dynamic contact process. At the time, the preload plays an important role in the contact boundary control and friction regulation of the stator and slider. Therefore, we supplied a nominal area coefficient cr based on the 2D bearing length ratio. When coupling it into the dynamic parameters modeling, the errors are reduced clearly. The error problem is critical for micromachines including smart motors and actuators, we will strive for the error gap in further study.
Round 3
Reviewer 1 Report
The introduction could be improved but I am clinging perhaps.
I do not have more questions.
Author Response
Q: The introduction could be improved but I am clinging perhaps.
A: Thanks for your comments. We have checked and refined the section of Introduction in the revised manuscript.
Reviewer 2 Report
1. Line 198, “when there is no sliding between the stator”, should it be “when there is no sliding between the stator and slider”?
2. Equations (5) and (18) are repeated.
3. Line 325, the meanings of the symbols in this equation should be explained.
4. In Figure 13, the calculation results in Fig. 13(a) is quite different from previous results, whereas the calculation results in Fig. 13(b) remains the same. Why? Does same value of Ca adopted in Fig. 13 (a) and (b)?
5. Manuscript should be polished by a native English speaker.
Author Response
Q1. Line 198, “when there is no sliding between the stator”, should it be “when there is no sliding between the stator and slider”?
A: Yes, thanks for your comment. We have moderated it in the revised manuscript.
Q2. Equations (5) and (18) are repeated.
A: Thanks. We have improved it in the revised manuscript.
Q3. Line 325, the meanings of the symbols in this equation should be explained.
A: We have supplied the explanation in the revised manuscript.
Q4. In Figure 13, the calculation results in Fig. 13(a) is quite different from previous results, whereas the calculation results in Fig. 13(b) remains the same. Why? Does same value of Ca adopted in Fig. 13 (a) and (b)?
A: Thanks for your comments. The effect of the applied voltage on the output speed is dominate and the resulted difference of different simulated models is not significant. Following the review’s comment, we have improved the simulated results in the revised manuscript with consideration of the nominal contact area ratio ca.
Q5. Manuscript should be polished by a native English speaker.
A: The revised manuscript has been checked by a native English speaker. We have added a necessary content in the section of Acknowledgments. Thanks.